# *Cimicifuga racemosa* Extract Ze 450 Re-Balances Energy Metabolism and Promotes Longevity

**DOI:** 10.3390/antiox10091432

**Published:** 2021-09-08

**Authors:** Malena Rabenau, Benjamin Dillberger, Madeline Günther, Sylvia Krippner, Veronika Butterweck, Georg Boonen, Jürgen Drewe, Gunter P. Eckert, Carsten Culmsee

**Affiliations:** 1Biochemical-Pharmacological Center (BPC), Institute of Pharmacology and Clinical Pharmacy, University of Marburg, Karl-von-Frisch-Strasse 2, 35043 Marburg, Germany; malena.rabenau@zellerag.ch (M.R.); madeline.guenther@staff.uni-marburg.de (M.G.); 2Center for Mind, Brain and Behavior, Hans-Meerwein-Strasse 6, 35032 Marburg, Germany; 3Preclinical Research, Max Zeller Soehne AG, 48590 Romanshorn, Switzerland; veronika.butterweck@zellerag.ch (V.B.); georg.boonen@zellerag.ch (G.B.); juergen.drewe@zellerag.ch (J.D.); 4Biomedical Research Center Seltersberg (BFS), Laboratory for Nutrition in Prevention and Therapy, Institute of Nutritional Sciences, Justus-Liebig-University of Giessen, Schubertstr. 81, 35392 Giessen, Germany; Dillberger@uni-giessen.de (B.D.); eckert@uni-giessen.de (G.P.E.); 5Biochemical-Pharmacological Center (BPC), Institute of Pharmacology, University of Marburg, Karl-von-Frisch-Strasse 2, 35043 Marburg, Germany; sylvia.krippner@gmail.com

**Keywords:** black cohosh, *Cimicifuga racemosa*, mitochondria, metformin, oxidative stress, longevity

## Abstract

Recently, we reported that the *Cimicifuga racemosa* extract Ze 450 mediated protection from oxidative cell damage through a metabolic shift from oxidative phosphorylation to glycolysis. Here, we investigated the molecular mechanisms underlying the effects of Ze 450 against ferroptosis in neuronal cells, with a particular focus on mitochondria. The effects of Ze 450 on respiratory complex activity and hallmarks of ferroptosis were studied in isolated mitochondria and in cultured neuronal cells, respectively. In addition, *Caenorhabditis elegans* served as a model organism to study mitochondrial damage and longevity in vivo. We found that Ze 450 directly inhibited complex I activity in mitochondria and enhanced the metabolic shift towards glycolysis via cMyc and HIF1α regulation. The protective effects against ferroptosis were mediated independently of estrogen receptor activation and were distinct from effects exerted by metformin. In vivo, Ze 450 protected *C. elegans* from the mitochondrial toxin paraquat and promoted longevity in a dose-dependent manner. In conclusion, Ze 450 mediated a metabolic shift to glycolysis via direct effects on mitochondria and altered cell signaling, thereby promoting sustained cellular resilience to oxidative stress in vitro and in vivo.

## 1. Introduction

*Cimicifuga racemosa* (L.) Nutt. (Ranunculaceae) (synonym of the accepted name *Actaea racemosa* L.) extracts have obtained a “well-established use” status in the treatment of menopausal complaints in the Committee of Herbal Medicinal Products conclusions included in the Monograph “Black Cohosh” of the European Medicines Agency (EMA/265439/2018). Although *Cimicifuga racemosa* (*C. racemosa*) extracts have been widely used for decades [1,2,3], their mechanisms of action remain to be elucidated. Climacteric symptoms include sweating, insomnia, mood changes, headache, vaginal dryness, and, most frequently, hot flushes [4]. It is well-known that during menopausal transition, the fluctuation of estrogen levels leads to alterations in central nervous system gene expression, neurotransmitter synthesis, and to an impairment of hypothalamic neuronal activity, including mitochondrial dysfunction and imbalanced redox homeostasis [5]. These alterations promote inflammatory processes and oxidative stress [6]. The aforementioned changes may largely contribute to the development of hot flushes by narrowing the thermoregulatory set point [7]. In line with the general view of menopausal complaints being caused by hormonal dysregulation, hormone replacement therapy (HRT) is commonly used to treat menopausal complaints [6], but has limitations due to adverse side effects, such as headache, uterine bleeding, and enhanced risks for venous thromboembolic disease, breast cancer, stroke, and coronary artery disease [8,9]. 

Menopausal transition and the corresponding symptoms, such as hot flushes, are related to an imbalance in redox homeostasis and are, therefore, often accompanied by the increased formation of reactive oxygen species (ROS) [10]. We recently reported that the *C. racemosa* extract Ze 450 mediates beneficial metabolic effects, protecting against oxidative cell damage in various cell types, including neuronal and liver cells [11,12]. Enhanced oxidative stress has been closely linked as a common pathological feature to many age-related disorders, metabolic diseases, and immunological impairments [13]. Additionally, Ze 450 demonstrated a strong influence on energy metabolism by reducing mitochondrial respiration and promoting glycolysis [12], similar to the Warburg effect [14]. To fulfill the energy demand of cells through glycolysis, larger amounts of glucose are required in comparison to the much more efficient oxidative phosphorylation (OXPHOS) in mitochondria.

In line with this, Ze 450 demonstrated antidiabetic effects in an *ob/ob* mouse model, as well as beneficial effects on metabolic parameters in ovariectomized rats, similar to the antidiabetic metformin [15,16]. Metformin is the first-line treatment option for most patients suffering from diabetes mellitus type II [17], and extremely interesting insights could be gained by comparing the effects of Ze 450 and metformin on mitochondria. Further, a recent retrospective observational study demonstrated that Ze 450 attenuated weight gain during menopause [18], confirming that Ze 450 may mediate metabolic effects in patients.

A great deal of research has been performed analyzing the Warburg effect, which describes the metabolic reprogramming towards aerobic glycolysis in cancer cells [14]. In these cells, such metabolic switches to glycolysis were associated with the activation of key transcriptional regulators like hypoxia-inducible factor α (HIF1α) and cMyc. *Cimicifuga racemosa* extract Ze 450 may exert similar regulation of energy metabolism through direct and indirect effects on mitochondrial OXPHOS.

In addition, former studies attributed the mode of action of *C. racemosa* extracts to an estrogen-like effect. Notably, this is a controversial discussion in the field, and the majority of more recent reports suggest that *C. racemosa* extracts should not be considered as phytoestrogens, and can neither activate estrogen receptors nor mediate estrogen hormone-like effects. In this study, Ze 450 was compared to 17β-estradiol, estrogen receptor agonists (4,4′,4″-(4-propyl-[1H]-pyrazole-1,3,5-triyl)trisphenol (PPT) and diarylpropionitrile (DPN)), and an estrogen receptor modulator (tamoxifen (TM)) to evaluate the similarities and differences of the extract to estrogen-mediated effects in model systems of oxidative stress and on mitochondrial parameters. A well-established in vitro model system of oxidative stress was used [19] to mimic the imbalance of redox homeostasis during the menopausal transition and emphasize the importance of metabolic changes besides estrogen depletion. Further, effects of Ze 450 and metformin were compared in vivo, in the model organism *Caenorhabditis elegans* (*C. elegans*) to assess the effects on longevity under basal conditions, and in conditions of oxidative stress exerted by the mitochondrial toxin paraquat.

## 2. Materials and Methods

### 2.1. Cimicifuga racemosa Extract Ze 450

The ethanolic (60% *v/v*) *C. racemosa* dry extract Ze 450 was manufactured from dried roots and rhizomes and obtained from Max Zeller and Soehne AG (Romanshorn, Switzerland), and thus, the indicated doses applied in this study contain 75% native extract. The content of triterpene glycosides was 6.4%. Ze 450 was dissolved in 60% ethanol (*v/v*) (Carl Roth GmbH, Karlsruhe, Germany) for all experiments. Ze 450 conforms to the herbal preparation B, which was granted with a well-established use status by a European Union herbal monograph on *C. racemosa* of the Herbal Medicinal Product Committee (HMPC) in 2010. The high-performance liquid chromatography (HPLC) fingerprint of the Ze 450 batch used (batch number: 151033) for the current experiments was published [12].

### 2.2. Reference Substances

Metformin hydrochloride (Cayman Chemical, Ann Arbor, MI, USA) was dissolved in H_2_O, 17β-estradiol (Merck KGaA, Darmstadt, Germany) in 60% ethanol and DPN, PPT and 4-OH-Tamoxifen (Merck KGaA, Darmstadt, Germany) were dissolved in DMSO, and all reagents were further diluted into culture medium to the indicated treatment concentrations, unless otherwise stated.

### 2.3. Cell Culture

HT22 cells (kindly provided by David Schubert, Cellular Neurobiology Laboratory, Salk Institute for Biological Studies, La Jolla, CA, USA) were grown in Dulbecco’s modified Eagle medium (DMEM, Capricorn Scientific GmbH, Ebsdorfergrund, Germany) supplemented with 10% heat-inactivated fetal calf serum (Merck KGaA, Darmstadt, Germany), 100 U/mL penicillin, 100 mg/mL streptomycin (Capricorn Scientific GmbH, Ebsdorfergrund, Germany) and 2 mM L-glutamine (Merck KGaA, Darmstadt, Germany). To induce cell death, erastin (Calbiochem^®^, Merck KGaA, Darmstadt, Germany) was added to the medium for the indicated amount of time (8–16 h).

### 2.4. Mitochondrial Isolation

Rat cortical mitochondria were isolated from Sprague-Dawley rats directly after dissection of the prefrontal cortex. All following steps were carried out on ice according to a well-established isolation method [20]. For cortical tissue, around 50 mg was minced with a scalpel in 1 mL of isolation buffer (300 mM sucrose, 5 mM 2-[[1,3-dihydroxy-2-(hydroxymethyl)propan-2-yl]amino]ethanesulfonic acid (TES), 200 µM ethylene glycol-bis(β-aminoethyl ether)-N,N,N′,N′-tetraacetic acid (EGTA), 1 mM dithiothreitol (DTT)). Cortical tissue was homogenized by trituration with a 20 G Neoject needle and strained through a 100 µm cell strainer (Life Science, VWR, Radnor, PA, USA). Afterwards, a pump-controlled cell homogenizer (Isobiotec, Heidelberg, Germany) with a constant pump flow rate (700 µL/min) was used to isolate mitochondria from the tissue. The system was primed with 1 mL mitochondrial isolation buffer before usage. Using the cell homogenizer, the cortical suspension was pumped three times with a clearance of 10 µm. Next, the system was rinsed with 1 mL mitochondrial isolation buffer to collect the total homogenate, followed by centrifugation at 800× *g* for 10 min at 4 °C (Heraeus™ FRESCO17, Thermo Fisher Scientific, Waltham, MA, USA). The supernatant was transferred to a new 1.5 mL tube and centrifuged at 9000× *g* for 10 min at 4 °C. The resulting pellet represents the crude mitochondrial fraction. After resuspension in 1 mL of mitochondrial isolation buffer (MSHE; 70 mM sucrose, 210 mM mannitol, 5 mM 4-(2-hydroxyethyl)-1-piperazineethanesulfonic acid (HEPES), 1 mM EGTA, 0.5% BSA) a BCA assay for protein determination using the Pierce BCA Protein Assay Kit (Thermo Fisher Scientific, Waltham, MA, USA) was performed. Protein concentration was calculated via linear regression with a BSA standard based on mitochondrial isolation buffer.

### 2.5. Cell Viability

Cell proliferation was analyzed in real-time by measuring electrical impedance [21]. Metabolic activity as an indicator of cell viability was quantified using the 3-(4,5-dimethylthiazol-2-yl)-2,5-diphenyltetrazolium bromide (MTT) assay [19]. Viable and metabolically active cells converted MTT (Merck KGaA, Darmstadt, Germany), which was added at a concentration of 2.5 mg/mL for 1 h at 37 °C to the culture medium, into purple formazan. Absorbance was measured at 570 nm vs. 630 nm with FluoStar (BMG Labtech, Ortenberg, Germany) after dissolving in DMSO (Carl Roth GmbH, Karlsruhe, Germany). Cell death of HT22 cells treated with Ze 450, erastin was detected using the Annexin-V-FITC/PI Detection Kit (PromoCell, Heidelberg, Germany) followed by fluorescence-activated cell-sorting (FACS, guava easyCyte, Merck KGaA, Darmstadt, Germany). Annexin-V-FITC was excited at 488 nm, and emission was detected through a 525 ± 30 nm bandpass filter. Propidium iodide (PI) was excited at 488 nm, and fluorescence emission was detected using a 690 ± 50 nm bandpass filter. Data were collected from at least 5000 cells with at least three replicates per condition.

### 2.6. Lipid Peroxidation

After the indicated treatments, HT22 cells were stained with BODIPY 581/591 C_11_ (Invitrogen, Carlsbad, CA, USA) for 1 h (37 °C, 4.5% CO_2_) and harvested for FACS analysis. Lipid peroxidation was analyzed by recording green (emission: 525 nm/30) and red (emission: 585 nm/50) fluorescence with a 488 nm excitation wavelength of at least 5000 cells of at least three replicates per condition. Levels of lipid peroxidation were calculated by the analysis of the shift from green to red fluorescence.

### 2.7. Mitochondrial ROS Formation

MitoSOX red (Invitrogen, Carlsbad, CA, USA) is selectively targeted to the mitochondria, where it is oxidized by superoxides exhibiting red fluorescence. For detection of mitochondrial ROS formation, MitoSOX red was applied for 30 min at 37 °C and cells were harvested for FACS analysis. Increasing red fluorescence correlating with the formation of mitochondrial ROS was detected by FACS analysis (excitation 488 nm, emission 690 nm/50). Data were collected from at least 5000 cells and three replicates per condition.

### 2.8. Mitochondrial Membrane Potential

After treatment with Ze 450 and erastin, cells were stained with tetramethylrhodamine, ethyl ester (TMRE) (0.4 nM for 30 min at 37 °C, MitoPT ΔΨm Kit, ImmunoChemistry Technologies, Bloomington, MN, USA) and harvested for TMRE fluorescence measurement via FACS analysis. Upon loss of the mitochondrial membrane integrity and, thus, membrane potential, a loss of TMRE fluorescence can be detected by FACS analysis (excitation 488 nm, emission 690 nm/50). Data were collected from at least 5000 cells and three wells per condition.

### 2.9. ATP Measurements

ATP levels were detected using the ViaLight™plus Kit (Lonza, Basel, Switzerland). After 24 h post-seeding in 96-well plates (6000 cells *per* well), cells were treated with Ze 450 and erastin. At the indicated time points after treatment, cells were transferred into a white 96-well plate and ATP levels were analyzed by luminescence detection with FluoStar OPTIMA (BMG Labtech, Ortenberg, Germany).

### 2.10. Seahorse Measurements on HT22 Cells

To determine the oxygen consumption rate (OCR) and extracellular acidification rate (ECAR) as measures of mitochondrial respiration and glycolysis, respectively, a simultaneous real-time measurement was performed using the XF Extracellular Flux Analyzer (Agilent Technologies, Santa Clara, CA, USA) as previously described [19]. Briefly, HT22 cells were plated in XF96-well microplates (6000 cells per well, Seahorse Bioscience, Agilent Technologies, Santa Clara, CA, USA) and treated with Ze 450 and erastin. At indicated treatment conditions, the growth medium was replaced by ~180 μL of assay medium (with 4.5 g/L glucose as the sugar source, 2 mM glutamine, 1 mM pyruvate, pH 7.35) and cells were incubated at 37 °C for 60 min. Three to six baseline measurements were recorded before adding the compounds. Oligomycin (ATP synthase inhibitor) (Merck KGaA, Darmstadt, Germany) was injected into port A (20 µL) at a final concentration of 3 µM, Carbonyl cyanide-4-(trifluoromethoxy)phenylhydrazone (FCCP) (uncoupling agent) (22.5 µL into port B) (Merck KGaA, Darmstadt, Germany) at a concentration of 0.5 µM, rotenone/antimycin A (complex I/III inhibitors) (25 µL into port C) (Merck KGaA, Darmstadt, Germany) at a concentration of 100 nM, and 1 µM 2-deoxyglucose (Carl Roth GmbH, Karlsruhe, Germany) (glycolysis inhibitor) at a concentration of 50 mM (27.5 µL into port D), respectively. Three measurements were performed after the addition of each compound (4 min mixing followed by 3 min detection).

### 2.11. Seahorse Measurement on Isolated Primary Cortical Mitochondria

#### 2.11.1. Electron Flow Assay

Electron flow assay was used to analyze the activity of mitochondrial complex I measuring the OCR with a Seahorse XFe69 Analyzer (Agilent Technologies, Santa Clara, CA, USA). After isolation of cortical mitochondria following established protocols [20], 10 µg of mitochondrial protein per well were re-suspended in 25 µL/well of mitochondrial assay solution (MAS; 70 mM sucrose, 220 mM mannitol, 10 mM KH_2_PO_4_, 5 mM MgCl_2_, 2 mM HEPES, 1 mM EGTA, 0.2% (*w/v*) BSA) with the addition of pyruvate (10 mM) and malate (2 mM). Subsequently, the plate was centrifuged at 2000× g for 20 min at 4 °C (Heraeus™ Megafuge 40R, Thermo Fisher Scientific, Hanau, Germany) to ensure adherence of the mitochondria to the bottom. Afterwards, 155 µL/well MAS was added to the final volume of 180 µL/well. Rotenone, succinate, and antimycin A were diluted in MAS without sucrose and mannitol, while ascorbate and N,N,N’,N’-tetramethyl-p-phenylenediamine (TMPD) were diluted in MAS and serially injected as follows: 2 µM rotenone (port A), 10 mM succinate (port B), 4 μM antimycin A (port C), 10 mM/100 μM ascorbate and TMPD (port D). For investigating complex IV-driven respiration, 1.0 M ascorbate in H_2_O (pH 7.2) and 10 mM TMPD in H_2_O (pH 7.2) were mixed with an equimolar concentration (10 mM) ascorbate to ensure TMPD remained reduced. One to three baseline measurements were recorded before adding the compounds, and one to three measurements were performed after the addition of each compound (3 min mixing followed by 3 min detection). Data analysis and visualization were performed using XFe Wave software 2.6.1 (Agilent Technologies, Santa Clara, CA, USA) and GraphPad Prism software 6.05 (Agilent Technologies, Santa Clara, CA, USA) displayed as point-to-point oxygen consumption rates.

#### 2.11.2. Coupling Assay

The coupling assay was used to measure the activity of mitochondrial complex II and III recording the OCR with a Seahorse XFe69 Analyzer (Agilent, Santa Clara, CA, USA). The coupling assay was performed as previously described with minor modifications [22]. Briefly, mitochondria were isolated as described before [23]. Briefly, 10 µg of cortical and 12 µg of liver mitochondrial protein per well were re-suspended in MAS with the addition of the complex II substrate succinate (10 mM) and the complex I inhibitor rotenone (2 μM). Subsequently, the plate was centrifuged at 2000× *g* for 20 min at 4 °C (Heraeus™ Megafuge 40R, Thermo Fisher Scientific, Hanau, Germany) to ensure adherence of the mitochondria to the bottom of the well. Afterwards, 155 µL/well MAS was added to the final volume of 180 µL/well. Modulators of the respiratory chain were diluted in MAS without sucrose and mannitol and serially injected as follows: 4 mM ADP (port A), 2.5 μg/mL oligomycin (port B), 4 μM FCCP (port C), 4 μM antimycin A (port D). One to three baseline measurements were recorded before adding the compounds, and one to three measurements were performed after the addition of each compound (3 min mixing followed by 3 min detection). Data analysis and visualization were performed using XFe Wave software 2.6.1 (Agilent Technologies, Santa Clara, CA, USA), and GraphPad Prism software 6.05 (GraphPad Software Inc., La Jolla, CA, USA), and displayed as point-to-point oxygen consumption rates.

### 2.12. Rhodamine Measurements

Rhodamine measurement was used to assess mitochondrial membrane potential integrity. Therefore, fluorescence quenching of rhodamine 123 (Thermo Fisher Scientific, Waltham, MA, USA) was measured over 40 min, as described previously [24]. Mitochondria were plated in a black 96-well plate with 75 µg mitochondrial protein per well and diluted in a buffer containing 200 mM sucrose, 10 mM MOPS-Tris, 5 mM succinate, 1 mM Pi(H_3_PO_4_), 10 µM EGTA, and 2 µM rotenone (pH 7.3). Loss of mitochondrial membrane potential was induced by FCCP (500 nM), which was injected at the end of the measurement. Fluorescence (excitation: 485 nm/12, emission: 530 nm/10) was measured with a FLUOstar OPTIMA reader (BMG Labtech, Ortenberg, Germany).

### 2.13. Glutathione Measurement

To determine glutathione (GSH) levels, HT-22 cells were seeded in 6-well plates (180,000 cells/well). After treatment with erastin for the indicated amount of time, two to three wells per condition were harvested by scratching, and washed on ice with PBS. GSH measurements were performed using the Glutathione Assay Kit (Cayman Chemical Company, Ann Arbor, MI, USA) following the manufacturer’s protocol. Briefly, cells were re-suspended in MES-buffer (0.4 M 2-(N-morpholino)ethanesulphonic acid, 0.1 M phosphate, 2 mM EDTA, pH 6.0) and homogenized by sonification. Insoluble fragments were removed by centrifugation at 10,000× *g* for 15 min. The supernatant was deproteinated by the addition of an equal volume of metaphosphoric acid (1.25 M). After incubation for 5 min, the mixture was centrifuged at 17,000× *g* for 10 min. Subsequently, the supernatant was mixed with a 4 M solution of triethanolamine to increase the pH. After transferral into a 96-well plate, the assay cocktail containing MES-buffer, co-factor mixture, enzyme mixture, and Ellman’s reagent was added. Absorbance was measured at 405 nm after 30 min of incubation. Total GSH amount was determined via standard curve calculation and normalized to the protein content.

### 2.14. Protein Analysis

Cells were lysed after the indicated time of treatment in 0.25 M D-mannitol, 0.05 M Tris base, 1 mM EDTA, 1 mM EGTA, 1 mM DTT, and 1% TritonX-100 supplemented with protease and phosphatase inhibitor cocktail tablets (Roche Diagnostics, Mannheim, Germany). Liquid nitrogen was used to disrupt cell membranes. Afterwards, the lysate was stored at −80 °C or thawed on ice and centrifuged for 15 min at 10,000× *g* at 4 °C. Protein lysate was used from the resulting supernatant. The total protein amount was determined using the Pierce BCA Protein Assay Kit (Thermo Fisher Scientific, Waltham, MA, USA) and 60 µg protein was loaded on a 7.5% polyacrylamide gel. After electrophoresis, the proteins were transferred from the gel onto a PVDF membrane (Roche Diagnostics, Basel, Switzerland). Then, the membranes were incubated with primary antibodies (overnight, at 4 °C), followed by incubation with the corresponding secondary antibody at room temperature. The membrane was incubated with luminol based HRP-Juice Plus (PJK GmbH, Kleinblittersdorf, Germany) for 2 min for protein detection. Visualization was realized with a ChemiDoc XRS system (Bio-Rad Laboratories Inc., Hercules, CA, USA). For densitometric protein analysis, the Bio Rad Image Lab Software was used. For loading control, a housekeeping protein (Actin, GAPDH, Vinculin) was detected in the same way. The antibodies were used as listed in Table 1.

### 2.15. Nematode and Bacterial Strain

*C. elegans* wild-type strain N2 was obtained from the *Caenorhabditis elegans* Genetics Center (University of Minnesota, Minneapolis, MN, USA). Nematodes were maintained on nematode growth medium (NGM) agar plates seeded with *E. coli* OP50 at 20 °C according to standard protocols [25]. For all experiments, synchronous populations were generated through a standard bleaching protocol [26].

### 2.16. Preparation of NGM Agar Plates

Three grams of NaCl, 2.5 g Bacto™ Peptone (Becton Dickinson, Heidelberg, Germany) and 17 g Bacto™ Agar (Becton Dickinson, Franklin Lakes, NJ, USA) were dissolved in 975 mL distilled water. After autoclaving, 1 mL cholesterol in ethanol (5 mg/mL), 1 mL 1 M CaCl_2_, 1 mL 1 M MgSO_4_, and 25 mL 1 M potassium phosphate buffer (pH 6.0) (108.3 g KH_2_PO_4_, 35.6 g K_2_HPO_4_, H_2_O to 1 L) were added in this order.

### 2.17. Cultivation and Treatment

Synchronous larvae were washed twice in M9 buffer (6 g Na_2_HPO_4_, 3 g KH_2_PO_4_, 5 g NaCl and 0.25 g MgSO_4_ × 7 H_2_O *per* L), counted and adjusted to 10 larvae per 10 µL. Nematodes were raised in cell culture flasks (Sarstedt, Nümbrecht, Germany) or OP50 spread NGM plates. OP50-NGM was added as a standardized food source with a volume 4.4-fold of the larvae containing M9 solution used. L1 larvae were maintained under shaking at 20 °C, reaching adulthood within 3 days. After reaching young adulthood, 48 h prior to the experiment, nematodes were treated with Ze 450 (100, 500, 1000 µg/mL) dissolved in EtOH 1% and metformin (25, 50, and 100 mM), or paraquat (5 mM; Merck KGaA, Darmstadt, Germany) dissolved in M9. Standard OP50, M9, and EtOH 1% served as controls.

### 2.18. Lifespan Assay

To determine the nematodes lifespan at 20 °C, a modified protocol from Amrit et al. [27] was applied and synchronized larvae, obtained from egg preparation as stated above, were raised on NGM agar plates spread with standard OP50 *E. coli* culture. After completing the L4 larval stage, 40 healthy animals per group were transferred to fresh NGM *E. coli* containing plates with a sterilized platinum wire. Effectors were incorporated into the OP50 culture with the concentration as needed. Nematodes were transferred to new plates every two days to distinguish between offspring until egg-laying stopped. In line with the separation from eggs and larvae, nematodes were checked for vital signs using a hot platinum wire held next to the animal’s heads. Worms showing no reaction to the heat stimulus were considered dead. The lifespan curves were statistically compared using the log-rank test.

### 2.19. Heat-Shock Survival Assay

Approximately 10 nematodes were raised per well in a 96-well microplate as mentioned above. After 48 h of incubation with effectors, time till death was determined using a microplate thermo-tolerance assay [28]. In brief, nematodes were washed off the wells with M9-buffer into 15 mL tubes followed by three additional washing steps. Each well of a black 384-well low-volume microtiter plate (Greiner Bio-One, Frickenhausen, Germany) was prefilled with 6.5 μL M9-buffer/Tween 20 (1% *v/v*). Subsequently, one nematode was immersed in 1 μL M9 buffer under a stereomicroscope (Breukhoven Microscope Systems, Capelle aan den Ijssel, The Netherlands). A volume of 7.5 μL SYTOX green (final concentration 1 μM; Life Technologies, Darmstadt, Germany), which penetrates only into cells with a compromised plasma membrane and gets fluorescent after binding to DNA, was added for fluorescent detection. To prevent water evaporation, the plates were sealed with a Rotilab sealing film (Greiner Bio-One, Frickenhausen, Germany). Heat shock (37 °C) was applied and fluorescence measured with a ClarioStar platereader (BMG Labtech, Ortenberg, Germany) every 30 min over the course of 17 h. The excitation wavelength was set at 485 nm, and the emission detected at 538 nm.

### 2.20. Chemotaxis Assay

Chemotaxis was assessed using a previously published method [29]. Briefly, agar plates were divided into four quadrants. Sodium azide (0.5 M) was mixed in the same parts with ethanol (95%) as control, or diacetyl (0.5%) as an attractant. Two microliters of either control or attractant solution were added to the center of two opposite quadrants with the same distance to the middle of the plate. Nematodes were washed and separated from larvae as stated above, and approximately 150 animals were placed in the center of the plate. After 1 h, each quadrant was counted, and a chemotaxis index calculated ((number of attractant–number of control)/number total).

### 2.21. Statistical Analysis

For statistical analysis of the survival curves, the log-rank test (Mantel-Cox test) was employed compared to the corresponding control. A *p*-value lower than 0.05 was considered to be a significant difference between two survival curves. All data are given as mean + or ±standard deviation (SD). Statistical comparison between treatment groups was performed by analysis of variance (two-way ANOVA) followed by Scheffé’s post hoc test, and a *p*-value < 0.05 was considered to be statistically significant. Calculations were executed with Winstat standard statistical software 2012.1.0.96 (R. Fitch Software, Bad Krozingen, Germany) and visualized using GraphPad Prism software 6.05 (GraphPad Software, San Diego, CA, USA).

## 3. Results

### 3.1. Antioxidant Properties of Ze 450 Do Not Predominantly Contribute to Its Protective Effects

As detected here in neuronal HT22 cells, Ze 450 neither affected glutathione levels nor rescued erastin-mediated glutathione depletion (Figure 1a). Ze 450 extract contains phenolic compounds, which may exert antioxidant properties. Therefore, it was a major aim of this study to investigate the potential radical scavenging activity of Ze 450 and to evaluate the corresponding impact on the observed protective effects. As detected in the DPPH assay, Ze 450 did indeed mediate moderate antioxidant activities, which were less pronounced in comparison to the radical scavenging activity of Trolox (Figure 1b).

Further, protein levels of enzymes involved in the antioxidant defense system were studied. Notably, catalase, copper/zinc superoxide dismutase (SOD), and manganese SOD protein levels remained unaltered in response to Ze 450 treatment (Appendix A). The question remained, however, whether the antioxidant properties of Ze 450 contributed to the observed protective effects in neurons [12]. Therefore, HT22 neuronal cells were challenged with erastin in the presence of the glycolysis inhibitor 2-deoxyglucose; 2-NDBG, (2-(N-(7-Nitrobenz-2-oxa-1,3-diazol-4-yl) amino)-2-deoxyglucose (2-DG). The results clearly show that Ze 450 is protective against erastin-mediated oxidative insult, but in the presence of 2-DG, this effect was abolished, while Trolox still mediated protection that was apparently independent of energy supply through glycolysis (Figure 1c). Based on these results, Ze 450 protects against oxidative cell death by reducing mitochondrial ROS formation and not by sustaining GSH levels (Figure 1a). Overall, these data demonstrate that the major mechanism underlying the protective effects of Ze 450 were not attributed to the moderate radical scavenger activity of Ze 450 nor any other antioxidant effect, but were dependent on the metabolic effects of the extract.

### 3.2. Ze 450 Directly Targets the Respiratory Chain Function in Isolated Mitochondria

In order to ascertain whether the effects of Ze 450 on mitochondria were mediated indirectly through cellular signaling pathways or were rather attributed to direct effects on mitochondrial respiration, we assessed the effects of Ze 450 in isolated cortical mitochondria from rats. The coupling assay was used to determine the activity of mitochondrial complex II and III in the presence of rotenone as an inhibitor for complex I and succinate as a substrate for complex II driven respiration. Ze 450 was exposed to isolated mitochondria for 30–60 min. As shown in Figure 2, Ze 450 mediated a time- and concentration-dependent reduction of mitochondrial OCR (Figure 2a,b). After 60 min, the respiration profile of mitochondria treated with 100 µg/mL Ze 450 was significantly reduced as revealed after injection of ADP and, moreover, complex II functionality was impaired, which was detected upon stimulation of full respiration by adding the ionophor FCCP (Figure 2b).

As described previously [20], Rhodamine 123 measurement was performed to assess mitochondrial membrane integrity as a quality control for isolated mitochondria. In this context, fluorescence quenching of all tested mitochondrial conditions was similar, indicating a good quality of mitochondria (Appendix A).

The electron flow through different complexes was analyzed with the initial presence of pyruvate and malate to assess mitochondrial complex I activity (Figure 2c). In line with previous findings [12], Ze 450 affected pyruvate and malate driven respiration in a time-dependent manner, but did not completely inhibit complex I activity, since the injection of rotenone (port A, Figure 2c) further reduced mitochondrial respiration. Injection of FCCP (port B, Figure 2c) led to an increase in OCR, proving that the electron flow was working properly. Further, it was excluded that the inhibitory effect of Ze 450 was attributed to decreased protein expression (Figure 2d), Appendix A). There were no differences detected between controls and Ze 450-treated cells regarding the expression levels of the proteins representing the mitochondrial respiratory complexes.

Taken together, Ze 450 exerted direct effects on mitochondria, as demonstrated by the inhibition of mitochondrial complex I-III activity in the isolated organelles in vitro.

### 3.3. Ze 450 Promotes a Glycolytic Phenotype

The major interest of this study was to gain new insights into the molecular mechanisms underlying the metabolic effects of Ze 450; therefore, we analyzed HIF1α and cMyc regulation, which were previously linked to metabolic switches to aerobic glycolysis. In the HT22 cells, Ze 450 regulated HIF1α protein levels in a time-dependent manner (Figure 3a,b). HIF1α protein levels were increased after 2 h of treatment with 100 μg/mL Ze 450 (Figure 3a,b) and returned to control level 16 h of Ze 450 treatment (Figure 3a,b). In addition to HIF1α, cMyc is known to be involved in metabolic reprogramming by promoting the rate of glycolysis, and besides its role as an oncogene, cMyc promotes important physiological functions, including energy metabolism [30]. cMyc protein levels were investigated in a time-dependent manner upon Ze 450 treatment (Figure 3c,d). Initially, there was no change detected (Figure 3c), but after 16 h of Ze 450 treatment, cMyc protein levels were significantly upregulated (Figure 3d).

These results demonstrate that Ze 450 alters HIF1α and cMyc protein levels in a consecutive time-dependent manner, with an early but transient HIF1α up-regulation followed by persistent cMyc regulation at later time points. HIF1α and cMyc are known to regulate key enzymes involved in glucose energy metabolism such as pyruvate dehydrogenase 1 (PDK1) and hexokinase II (HXKII) [30,31]. Ze 450 led to an increase in PDK1 and HXKII protein levels after 16 h of treatment with Ze 450 (Figure 3e–g). Additionally, glucose uptake was monitored by measuring 2-NBDG fluorescence. The results show that Ze 450 promoted 2-NBDG uptake compared to control conditions (Appendix A and Figure 4a–d).

Overall, these results suggest a link between Ze 450 and key regulators of energy metabolism such as HIFα and cMyc, thereby, enhancing glucose uptake and metabolic reprogramming through accelerating key enzymes of glycolysis with concomitantly reduced OXPHOS in mitochondria.

### 3.4. Effects of Estrogen, Estrogen Agonists and 4-OH-Tamoxifen on Neuroprotection

Previously, the therapeutic effects of *C. racemosa* extracts were attributed to phytoestrogens, although classic phytoestrogens were not identified as major components of the extract and according to effects on estrogen receptors were not confirmed. Here, we compared the effects of 17β-estradiol, estrogen receptor agonists and 4-OH-tamoxifen to Ze 450-mediated effects in HT22 neuronal cells. Under basal culture conditions, only very high concentrations of 100 µM 17β-estradiol reduced metabolic activity as detected in the MTT assay (Figure 4a), while the erastin-mediated cytotoxic reduction of the metabolic activity was prevented at 50 µM of 17β-estradiol (Figure 4a). These results were confirmed in the annexinV/PI cell death assay showing that 17β-estradiol preserved cell viability at 50 µM (Figure 4b,c), while lower concentrations of the estrogen derivate did not provide a protective effect (Figure 4b,c). In addition, the BODIPY FACS analysis revealed a concentration-dependent protection by 17β-estradiol at the level of lipid peroxidation (Figure 4d). In conclusion, these results suggested that estradiol-mediated protection against oxidative damage was independent of estrogen-receptor-mediated effects, since only high micro molar concentrations of 17β-estradiol mediated protection, whereas concentrations of the estrogen relevant for receptor stimulation showed no effect.

In addition, mitochondrial respiration of cells treated with 17β-estradiol was compared to the effects of Ze 450 (Appendix A). In contrast to Ze 450, 1 µM 17β-estradiol led to an increase in mitochondrial respiration (Appendix A). To analyze effects independent of basal OCR differences, the spare respiratory capacity was analyzed (Appendix A). The results showed that Ze 450 and 50 µM 17β-estradiol reduced spare respiratory capacities compared to control conditions (Appendix A). In contrast to Ze 450, erastin-mediated reduction of OCR was partly prevented by 50 µM 17β-estradiol (Appendix A). Analyzing the rate of glycolysis, the reduction of the ECAR mediated by erastin was partly prevented by 50 µM 17β-estradiol and Ze 450 (Appendix A).

To further elucidate the potential role of estrogen receptor stimulation, estrogen receptor agonists (DPN and PPT) were applied in the model system of erastin-induced oxidative death. DPN and PPT differ in their binding affinity towards estrogen receptors (ER), with DPN preferentially binding to ERα and PPT binding to ERβ. Metabolic activity was not affected by any of the ER agonists, and they did not prevent erastin-mediated oxidative damage, assessed by MTT assay (Appendix A). Furthermore, the combined treatment of either PPT or DPN with 100 µg/mL of Ze 450 did not exceed Ze 450-mediated effects (Appendix A).

Moreover, 4-OH-tamoxifen, a selective estrogen receptor modulator was tested. Tamoxifen is a known ER antagonist clinically used to treat estrogen-dependent breast cancer [32]. Here, the effects of 4-OH-tamoxifen were compared to Ze 450 in neuronal cells. At 50 µM, 4-OH-tamoxifen exerted strong reduction of metabolic activity, likely reflecting toxic effects and cell death at this high concentration (Appendix A), while 10 µM 4-OH-tamoxifen was protective against erastin-mediated oxidative damage (Appendix A). To further study synergistic or additive effects of Ze 450 with 4-OH-tamoxifen, low, non-protective concentrations of 4-OH-tamoxifen (0.5 and 1 µM) were used (Appendix A). The results show that no additional effects were detected in comparison to Ze 450 (Appendix A). Supporting the previous findings, Ze 450 protected against erastin-mediated cell death, while there was no additive or synergistic effect observed either for ER agonists (DPN, PPT) or for 4-OH-Tamoxifen (Appendix A).

In summary, these findings demonstrated that estrogen and 4-OH tamoxifen-mediated effects were independent of estrogen receptor signaling, and, most importantly, Ze 450-mediated effects were not attributed to estrogen-like effects.

### 3.5. Metformin Mediates Protective Effects on the Level of Mitochondria

In the present study, the effects of metformin were compared to Ze 450 in a model system of oxidative stress to elucidate similarities or differences in their mechanism of action.

Metformin did not affect metabolic activity in the tested concentration range as assessed by the MTT assay under basal culture conditions (Figure 5a). In the model of erastin-meditated oxidative death, 50 mM metformin preserved cell viability assessed by the MTT assay comparable to 100 µg/mL of Ze 450 (Figure 5a). Annexin V/PI FACS measurements confirmed a concentration-dependent protection against erastin-induced cytotoxicity by metformin (Figure 5b).

Ze 450 has been demonstrated to exert strong protective effects on lipid-peroxidation in model systems of oxidative stress [11,12]. Hence, metformin-mediated effects on cellular ROS formation were investigated. The results showed that metformin was capable of mediating protection against erastin-mediated increase in lipid-peroxidation, but this effect was less pronounced compared to 100 µg/mL Ze 450 (Appendix A). Furthermore, mitochondrial ROS formation was analyzed using MitoSOX staining and subsequent FACS analysis. Erastin-induced increase in the production of mitochondrial superoxides was reduced upon concentrations greater than 2 mM metformin (Appendix A). In line with this, metformin protected against erastin-mediated loss of mitochondrial membrane potential in a concentration-dependent manner (Appendix A). In contrast to Ze 450, metformin decreased the ATP levels in the cells (Appendix A). Moreover, erastin-induced ATP depletion was completely prevented by 100 µg/mL Ze 450, whereas metformin merely affected the erastin-induced decline of ATP levels (Appendix A).

Further, we evaluated the influence of metformin on energy metabolism (Figure 6a–d). Ze 450 showed a concentration-dependent reduction of OCR, which represents mitochondrial respiration, and this effect was also detectable for metformin (Figure 6a). Metformin (5 and 10 mM) decreased basal and FCCP stimulated OCR in a comparable manner as Ze 450 (*p* < 0.001 compared to control cells), whereas the effect with 2 mM Metformin was less pronounced for FCCP-stimulated OCR (*p* < 0.05) (Figure 6a). Erastin-mediated reduction of ECAR was prevented by Ze 450, whereas metformin did not rescue glycolytic energy supply to the same extent (Figure 6d).

Interestingly, metformin and Ze 450 provided comparable protective effects in the model system of oxidative cell death, but regarding effects on the energy metabolism, the effects of metformin were less pronounced compared to Ze 450.

### 3.6. Metformin and Ze 450 Extend Lifespan in C. elegans

*C. elegans* is a well-established model system to study longevity and mitochondrial stress resilience upon paraquat (PQ) exposure in vivo. The aim of this study was to investigate the potential protective effects of Ze 450 against paraquat-induced death and, further, evaluating the impact of Ze 450 on longevity in comparison to metformin. First, the heat shock assay was used as a pre-test system to define a concentration range for Ze 450 and metformin that should then be applied in long-term lifespan assay (Figure 7a,b). Based on the heat shock assay, 100 µg/mL Ze 450 was chosen as an appropriate concentration for the PQ stress test. In the PQ assay, Ze 450 enhanced overall survival and improved median survival (median survival: Ze 450 9.5 h) compared to survival in *C. elegans* exposed to paraquat (median survival: PQ 8 h) (Figure 7c).

It has been demonstrated that metformin prolongs the lifespan in mice [33] and in *C. elegans* [34,35]. Since similar mechanisms may cause the effects of Ze 450 and metformin, metformin was used as a reference condition. In the heat shock assay, 50 and 100 mM metformin decreased survival, while 25 mM led to an increase in survival compared to the vehicle controls (Figure 7b). Additionally, 25 mM metformin improved the survival in PQ-challenged nematodes compared to PQ-exposed controls (Figure 7d).

Following these promising findings, Ze 450 and metformin were further investigated for their effects on the lifespan of *C. elegans* in a real-time assay at concentrations matching concentrations previously used in the cultured neuronal cells (Figure 7e,f). The results showed that Ze 450 prolonged the median survival (24 days) and the overall survival (Figure 7e) compared to vehicle control (20 days). Moreover, 25 mM metformin demonstrated similar results regarding the overall survival rate compared to vehicle control (Figure 7d), but metformin did not improve median survival (22 days M9 and 25 mM metformin). In this real-time setting, we demonstrated robust effects of Ze 450 on the lifespan, which were more pronounced compared to metformin.

Besides the studies on longevity, *C. elegans* is used to investigate neuronal function, based on the nematodes’ well-developed sensory and olfactory system. Chemotactic behavior was studied to assess whether Ze 450 affects neuronal function (Appendix A). Metformin and Ze 450 increased the chemotactic index compared to the corresponding control (Appendix A).

Taken together, the findings in *C. elegans* suggest that Ze 450 can mediate beneficial effects regarding neuronal function and longevity in a whole organism in vivo.

## 4. Discussion

Although *C. racemosa* extracts are widely used to treat menopausal complaints, the exact mechanism of action has not been clarified. The physiological process of menopausal transition is accompanied not only by estrogen depletion but also by age-related metabolic alterations, including for example an increase in body weight. Consequently, meno-pausal women are exposed to enhanced risk factors for metabolic diseases, such as diabetes mellitus type II or osteoporosis [10,36]. Ze 450 was recently investigated for its impact on metabolic parameters of menopausal women [18]. The authors demonstrated that Ze 450 attenuated weight gain in menopausal women. Additionally, it was shown that Ze 450 mediated beneficial effects, including reduction in body weight and improved glucose metabolism in a mouse model of diabetes [16]. In the present study, we revealed that the *C. racemosa* extract Ze 450 mediated advantageous effects on cellular energy metabolism and according to protective effects against oxidative cell damage that were independent of estrogen signaling. In this context, it has been shown that *C. racemosa* extract did not affect hormone levels in peri- and postmenopausal women [37].

Menopausal complaints are frequently treated with HRT, although this has some limitations due to adverse side effects, such as coronary heart disease, stroke, or increased risk of developing breast cancer. In this study, it was demonstrated that the mechanism of action of Ze 450 is distinct from the effects of 17β-estradiol, estrogen receptor agonists DPN and PPT, and of the SERM tamoxifen (Figure 4, supporting Appendix A). We found that 17β-estradiol protected against erastin-mediated damage, but this effect was attributed rather to its antioxidant properties [38,39,40]. Additionally, mitochondrial respiration was slightly increased by 1 µM of 17β-estradiol compared to control conditions (Appendix A). This is in sharp contrast to the effects of Ze 450, which decreased OCR as a mechanistic basis of the observed protective effects.

Moreover, anti-proliferative effects have been demonstrated in vitro [12], and *C. race-mosa* extract has been proposed to mediate protective effects in cancer patients [41,42,43]. Taken together, the findings obtained in the current study indicate a mechanism of action of Ze 450, which is distinct from estrogen-mediated signaling, underlining the importance of *C. racemosa* extracts as hormone-free alternatives to treat menopausal complaints, and further suggesting a beneficial influence on metabolic disorders [15,16].

Our previous studies [12] and the current results demonstrated that Ze 450 reduced mitochondrial respiration in neuronal cells and promoted an inhibitory effect on mitochondrial complexes in a concentration- and time-dependent manner. Such inhibition of mitochondrial respiration due to reduced mitochondrial complex functionality leads to reduced superoxide production; thus, supporting the hypothesis that Ze 450 mediates protective effects by reducing mitochondrial ROS formation. Consequently, the major production sites of superoxides in mitochondria are inhibited by Ze 450, which subsequently reduces overall oxidative damage to the cells. Our current results point at partial inhibition of CI by Ze 450, but also direct or indirect inhibition of other respiratory complex activities may be involved. So far, we could not detect inhibition of CII or CIII by Ze 450, and, therefore, focused on other mechanisms that may cause the observed metabolic shift from OXPHOS to glycolysis.

A similar glycolytic switch detected here in neuronal cells exposed to Ze 450 was first described as a fundamental mechanism in cancer cells, and was termed as the Warburg effect. The underlying molecular mechanisms of the Warburg effect are not fully understood, but HIF1α and cMyc have been recognized as master regulators of important glycolytic enzymes like HXKII and PDK1, promoting metabolic reprogramming. In the present model system in neuronal cells, Ze 450 regulated HIF1α and cMyc protein levels in a shifted time dependency, which finally led to an enhanced glycolytic metabolism, by increasing HXKII and PDK1 protein levels.

HIF has been described to be involved in TCA regulation by enhancing PDK1 activity and thereby limiting the entry of pyruvate into the TCA. Consequently, mitochondrial respiration is decreased [44,45]. In order to examine the impact on HIF-mediated regulation of energy metabolism, PDK1 protein levels were measured. Concomitant with HIF up-regulation, Ze 450 increased PDK1 protein levels, and mitochondrial respiration measured through OCR was decreased [12]. Moreover, our results in isolated mitochondria suggested that this effect on OCR was accelerated by a direct inhibition of mitochondrial respiratory chain complexes by Ze 450.

Further, it was demonstrated that HIF is involved in the alteration of COX4 subunits in response to hypoxia in mammalian cells by restoring the electron transfer to O_2_ [46]. This consequently leads to reduced superoxide formation by CI and CIII. In this regard, the regulation of superoxide formation seems to be a crucial metabolic adaption regulated by HIF [46,47], and this effect may also contribute to the observed protective effects by Ze 450.

Furthermore, imbalanced redox homeostasis has been linked to many pathological conditions. For example, hot flushes and increased ROS formation have been linked to the metabolic impairments in diabetic patients [48]. We showed previously that Ze 450 reduced oxidative stress-mediated mitochondrial impairment and cell death in hypothalamic cells [12] thereby, proposing a potential link to the reported metabolic effects of the *C. racemosa* extract; however, the mechanism of action of Ze 450 in experimental settings in vivo and in patients is still unknown. Besides the effective treatment of menopausal complaints, it was suggested that Ze 450 has further therapeutic potential in metabolic diseases. For example, Ze 450 mediated antidiabetic effects in an ob/ob mouse model, which were comparable to metformin [16], highlighting the potential for Ze 450 as a possible early treatment option for metabolic diseases, such as type II diabetes. This was further supported by clinical data demonstrating reduced weight gain in postmenopausal women taking Ze 450 [18].

Obesity is one of the most important risk factors for type II diabetes, and the first line of treatment of the disease with the non-insulinotropic antidiabetic drug metformin is known for attenuating body weight gain. Therefore, in the present study, it was of major interest to compare the mode of action of Ze 450 with metformin. Metformin regulates blood glucose levels and fasting insulin levels by reducing gastrointestinal and increasing peripheral glucose uptake [17,49,50]. So far, however, the molecular mechanism of action of metformin requires clarification. It has been suggested that metformin acts at the cellular level as a mitochondrial ETC inhibitor and an activator of AMP-activated protein kinase (AMPK) [50], thereby interfering with several important regulation pathways of energy metabolism [51]. Moreover, AMPK activation has been related to neuroprotective effects [52]. Both metformin and Ze 450 protected against mitochondrial ROS formation, loss of mitochondrial membrane potential and oxidative cell death. Additionally, metformin has been described as a mitochondrial CI inhibitor leading to reduced mitochondrial ROS production [50,51]. This was also found in the present study, since metformin reduced OCR in a concentration-dependent manner. Here, we found that Ze 450 demonstrated stronger protective effects on the rescue of erastin-mediated ATP depletion compared to metformin. This might be linked to a more pronounced shift towards glycolysis under oxidative stress conditions mediated by Ze 450. These results demonstrate that Ze 450 can interfere with cellular energy metabolism and thereby mediate beneficial effects, which could expand its therapeutic options.

Therefore, it was of major interest to compare the effects of metformin and Ze 450 on a whole organism. For that purpose *C. elegans* was used as a suitable in vivo system to evaluate effects on mitochondrial related oxidative stress and longevity [53]. In the present study, 100 μg/mL of Ze 450 protected *C. elegans* against PQ-mediated toxicity. Differences between metformin and Ze 450 were identified, since 25 mM metformin enhanced survival in the heat-stress resistance test, while 100 μg/mL of Ze 450 did not. In previous studies, the lifespan extension in *C. elegans* by metformin was attributed to a multitude of effects, for example, the modulation of the microbial folate and methionine metabolism [35], and by activation of AMPK and lysosomal pathways [34]. Further, metformin-mediated effects on aging are investigated in ongoing clinical trials [54]. This study may deepen the knowledge about other therapeutic indications beyond its impact on diabetes mellitus type II [55]. Similarly, the underlying mechanisms promoting longevity and protection against oxidative challenge by *C. racemosa* extract in vivo warrant further investigation, also in clinical settings.

The results obtained with *C. elegans* in vivo emphasizes that Ze 450 has beneficial effects on the whole organism and might contribute to healthy aging. Further, Ze 450 demonstrated protective effects against the oxidative challenge by PQ, confirming the previous findings in vitro, which showed that Ze 450 preserved mitochondrial function and integrity in paradigms of ferroptosis.

## 5. Conclusions

In summary, Ze 450 directly affects mitochondria and, thereby, enhanced sustained cellular resilience to oxidative stress. Overall, these findings support the conclusion that Ze 450 has a strong impact at the level of mitochondria, which was demonstrated in isolated mitochondria, in neuronal cells in vitro, and in *C. elegans* in vivo. Our results provide evidence that Ze 450 has additional beneficial effects on age-related and metabolic diseases.

## Figures and Tables

**Figure 1 antioxidants-10-01432-f001:**
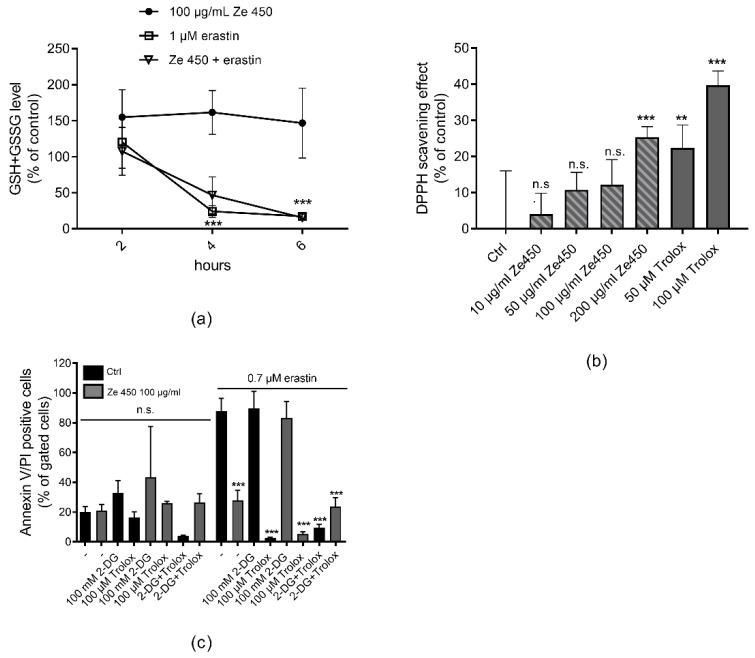
Protective effects of Ze 450 are independent of its antioxidant properties. (**a**) Effects of Ze 450 on cellular glutathione levels (*n* = 5–7/treatment condition). Data are shown as mean + SD (*** *p* < 0.001 compared to untreated control; ANOVA, Scheffé’s test). (**b**) DPPH radical scavenging activity was determined at 517 nm. (*** *p* < 0.001, ** *p* < 0.01 compared to control; ANOVA Scheffé’s test; n.s. not statistically significant). (**c**) Cell death was assessed by flow cytometry using Annexin V and PI after 16 h of treatment in HT22 cells. The bar graph shows one representative experiment with three replicates per sample (mean + SD; 5,000 cells per replicate; *** *p* < 0.001 compared to erastin treated control; ANOVA Scheffé’s test).

**Figure 2 antioxidants-10-01432-f002:**
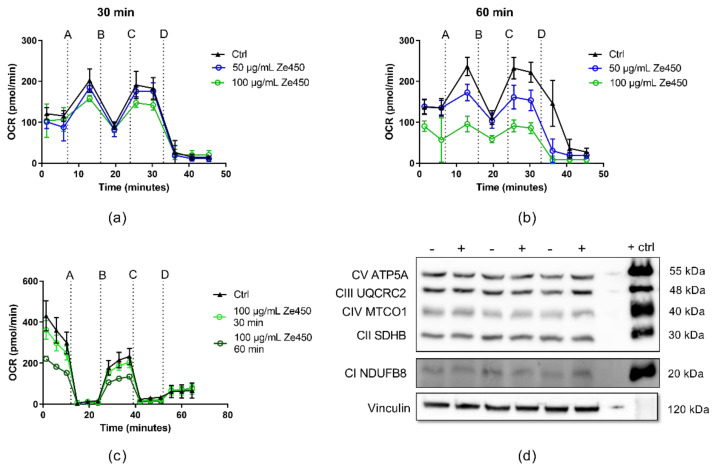
Ze 450 reduces mitochondrial respiration. (**a**,**b**) Oxygen consumption rate (OCR) was analyzed after 30 and 60 min of incubation with Ze 450 (50 and 100 µg/mL). ADP (port A), oligomycin (port B), FCCP (port C) and antimycin A (port D) were injected, successively. (**c**) OCR was analyzed after 30 and 60 min of incubation with Ze 450 (50 and 100 µg/mL). Rotenone (port A), FCCP (port B), antimycin A (port C) and ascorbate/TMPD (port D) were injected, successively. (**d**) depicts an immunoblot. Relative protein levels of the five electron transport chain complexes (CI-CV) were assessed in HT22 cells treated with 100 μg/mL Ze 450 for 16 h compared to untreated control (− control, + Ze 450). Rat cortical mitochondria served as a positive control (+ ctrl); ANOVA, Scheffé’s test.

**Figure 3 antioxidants-10-01432-f003:**
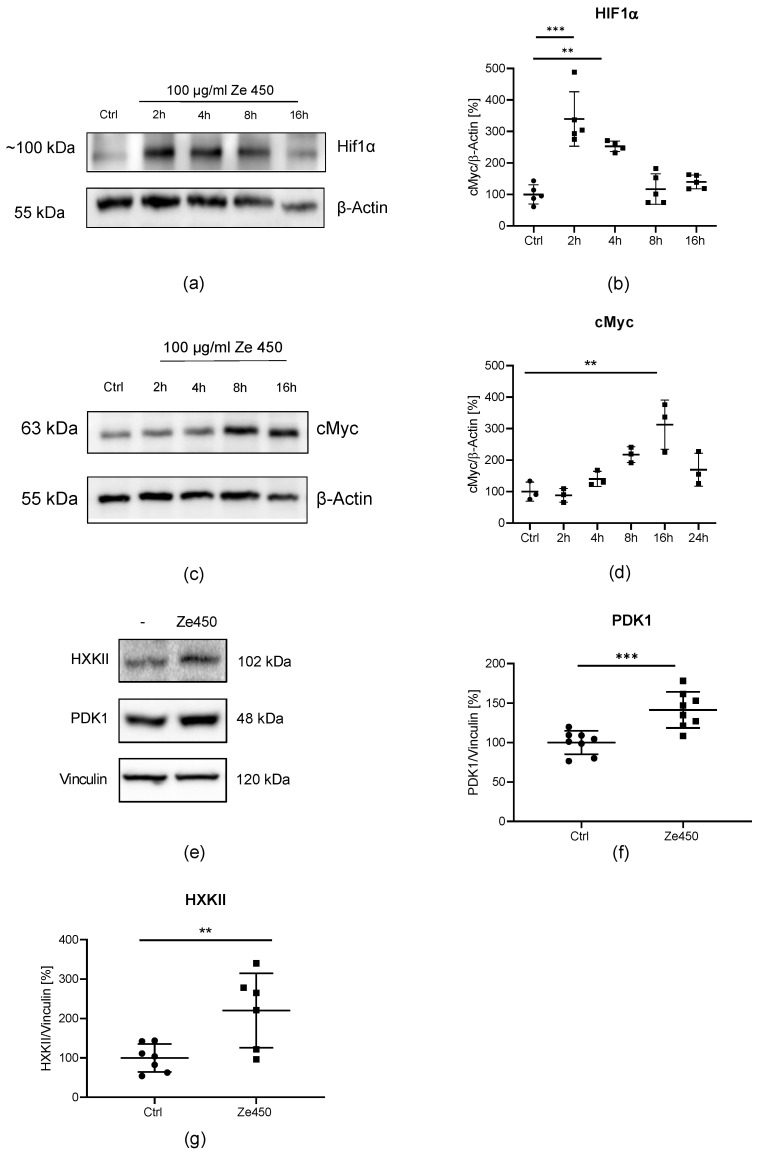
Effects of Ze 450 on protein levels of metabolic key regulators. (**a**) Effects of 100 μg/mL Ze 450 on HIF1α protein levels were investigated via Western Blot after 2–16 h treatment in HT22 cells. (**b**) Quantification of Western Blot after 2–16 h of treatment with 100 µg/mL Ze 450 depicted as optical density compared to β-Actin as loading control (*n* = 5/treatment condition; *** *p* < 0.001; ** *p* < 0.01; unpaired *t*-test). (**c**) Representative Western Blot showed the effect of 100 μg/mL Ze 450 treatment on cMyc protein level after 2–16 h treatment in HT22 cells. (**d**) Quantification of Western Blot as optical density compared to β-Actin as loading control (*n* = 3/treatment condition; ** *p* < 0.01; unpaired *t*-test). (**e**) Representative Western Blot of PDK1, and HXKII protein levels after 16 h of Ze 450 treatment (100 μg/mL) in HT22 cells. (**f**,**g**) Quantification of Western Blot as optical density compared to vinculin as loading control (*n* = 8/treatment condition, *** *p* < 0.001; ** *p* < 0.01, unpaired *t*-test).

**Figure 4 antioxidants-10-01432-f004:**
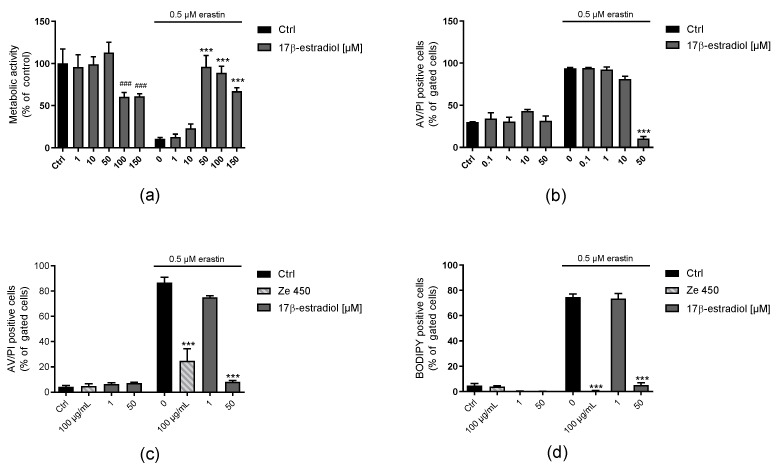
Effects of 17β-estradiol on HT22 cells. (**a**) Metabolic activity was evaluated in erastin-challenged HT22 cells by measuring MTT formazan absorbance (0.5 μM, 16 h). Eight replicate wells per condition are displayed as mean + SD. The absorbance under control conditions is set as 100%. (**b**,**c**) Cell death was assessed by flow cytometry using Annexin V and PI staining after 16 h of treatment in HT22 cells. The bar graph shows one representative experiment with three replicates per sample (mean + SD; 5000 cells per replicate). (**d**) Lipid-peroxidation was assessed by flow cytometry using BODIPY™ after 8 h. The bar graph shows one representative experiment with three replicates per sample (*** *p* < 0.001, compared to erastin-treated control; ### *p* < 0.001 compared to untreated control; ANOVA, Scheffé’s test).

**Figure 5 antioxidants-10-01432-f005:**
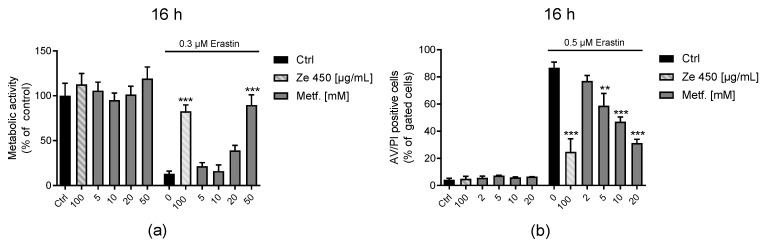
Metformin protects HT22 cells against oxidative damage. (**a**) Metabolic activity was evaluated in erastin-challenged HT22 cells by measuring MTT formazan absorbance (0.3 μM, 16 h). Eight replicate wells per condition are displayed as mean + SD. The absorbance under control conditions is set as 100%. (**b**) Cell death was assessed by flow cytometry using Annexin V and PI after 16 h of treatment in HT22 cells. The bar graph shows one representative experiment with three replicates per sample (mean + SD; 5000 cells per replicate). (**a**,**b**) *** *p* < 0.001, ** *p* < 0.01 compared to erastin treated control (ANOVA, Scheffé’s test).

**Figure 6 antioxidants-10-01432-f006:**
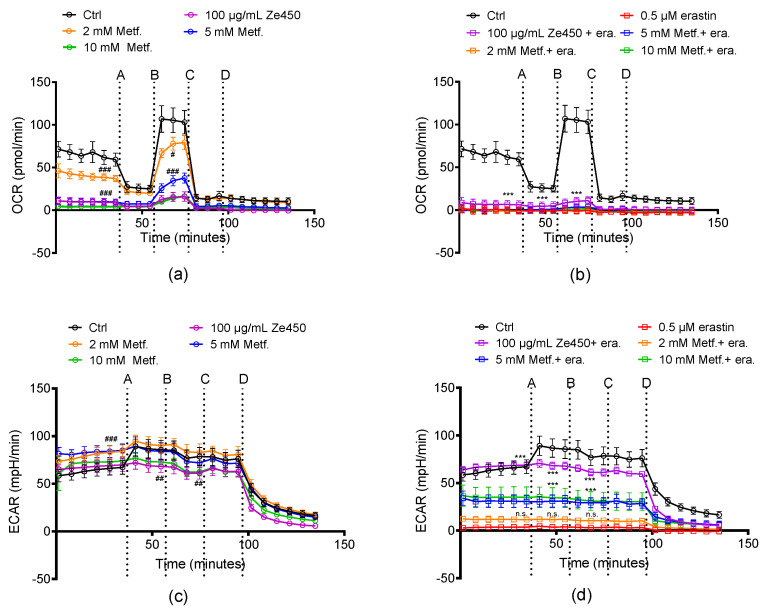
Metformin mediated effects on OCR and ECAR. Data of a representative experiment is depicted, showing OCR and ECAR under basal and oxidative stress conditions. (**a**,**b**) After 16 h of Ze 450 and metformin treatment (**a**) and challenged with 0.5 μM erastin (**b**), the oxygen consumption rate (OCR) of HT22 was analyzed with a Seahorse XFe96 Analyzer. Data of 5–8 replicate wells per condition are given as mean ± SD. C, D: After 16 h of treatment with Ze 450 and metformin (**c**) and challenged with 0.5 μM erastin (**d**) the extracellular acidification rate (ECAR) was determined with a Seahorse XFe96 Analyzer. Data of 5–8 replicate wells per condition are given as mean ± SD (port A: oligomycin; port B: FCCP; port C: rotenone and antimycin A; port D: 2-deoxyglucose; *** *p* < 0.001, compared to erastin-treated control; ### *p* < 0.001; ## *p* < 0.01; # *p* < 0.05 compared to untreated control; ANOVA, Scheffé’s test; n.s. not statistically significant).

**Figure 7 antioxidants-10-01432-f007:**
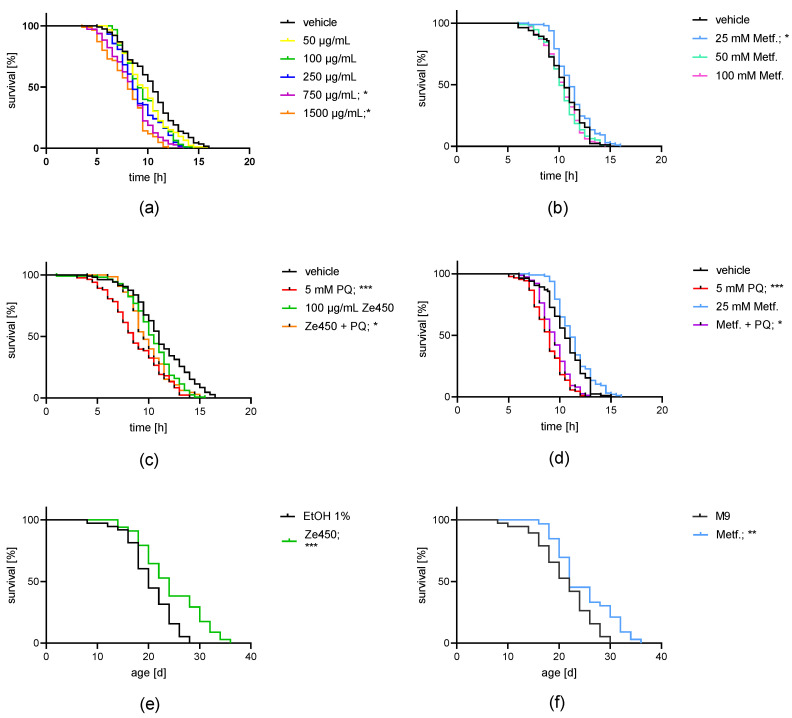
Effects of Ze 450 and metformin on *C. elegans*. (**a**) Effects of different concentrations of Ze 450 in a heat stress model system in *C. elegans*. The 50–250 μg/mL Ze 450 did not affect the lifespan, while worms treated with 750 or 1500 μg/mL showed a decrease in lifespan (* *p* < 0.05, *n* = 85–115). (**b**) Effects of metformin (25–100 mM) in a heat stress model system in *C. elegans*. 25 mM metformin improved lifespan (* *p* < 0.05 compared to M9 vehicle control), while treatment with ≥50 mM metformin did not affect lifespan (*n* = 75–97). (**c**) Effects of 100 μg/mL Ze 450 in a model system of paraquat (PQ) induced oxidative stress in *C. elegans* (*n* = 65–113, *** *p* < 0.001 PQ compared to vehicle, * *p* < 0.05 Ze 450 + PQ compared to PQ). (**d**) Effects of 25 mM metformin in a model system of paraquat (PQ)-induced oxidative stress in *C. elegans* (*n* = 76–97, *** *p* < 0.001 PQ compared to M9 vehicle, * *p* < 0.05 25 mM metformin +PQ compared to PQ). (**e**) Effects of 100 µg/mL Ze 450 on *C. elegans* in the real-time lifespan assay (*** *p* < 0.001 compared to vehicle; 1% ethanol). (**f**) Effects of 25 mM metformin on *C. elegans* in the real-time lifespan assay (** *p* < 0.001 compared to vehicle; M9).

**Table 1 antioxidants-10-01432-t001:** Antibodies used for Western Blot.

Antibody	Company	Host	Dilution
Catalase	Santa Cruz, Dallas, Texas, USA	Mouse	1:1000 in 5% milk
cmyc	Cell signaling, Danvers, MA, USA	Rabbit	1:1000 in 5% milk
CuZnSOD	Santa Cruz, Dallas, Texas, USA	Mouse	1:1000 in 5% milk
HIF1α	Cell signaling, Danvers, MA, USA	Rabbit	1:1000 in 5% milk
HXKII	Cell signaling, Danvers, MA, USA	Rabbit	1:1000 in 5% milk
mitoProfile OXPHOS	Abcam, Cambridge, MA, USA	Mouse	1:250 in 5% milk
MnSOD	Sigma Aldrich, Burlington, MA, USA	Rabbit	1:500 in 5% milk
PDK1	Enzo Life Science, Farmingdale, NY, USA	Rabbit	1:1000 in 5% milk
pmTOR	Cell signaling, Danvers, MA, USA	Rabbit	1:1000 in 5% milk
β-Actin	Novus Biologicals, Centennial, CO, USA	Mouse	1:2000 in 5% milk
GAPDH	Cell signaling, Danvers, MA, USA	Rabbit	1:50,000 in 5% milk
Vinculin	Sigma-Aldrich, Darmstadt, Germany	Mouse	1:10,000 in 5% milk

## Data Availability

Some of the data presented in this study are openly available at the publication server of the Philipps University of Marburg (https://archiv.ub.uni-marburg.de/ubfind/Record/urn:nbn:de:hebis:04-z2020-0055 (accessed on 19 December 2019)) in: Malena Rabenau. Targeting mitochondria: Novel concepts for mechanisms underlying therapeutic effects of Cimicifuga racemosa and Petasites hybridus extracts. https://doi.org/10.17192/z2020.0055 (accessed on 19 December 2019).

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
