# Peer review of "Cimicifuga racemosa Extract Ze 450 Re-Balances Energy Metabolism and Promotes Longevity"

_antioxidants, 2021, doi:10.3390/antiox10091432_

Round 1

Reviewer 1 Report

The manuscript, “Cimicifuga racemosa Extract Ze 450 Re-balances Energy Metabolism and Promotes Longevity”, compared the effects of Ze 450 to other well-studied chemicals and drugs and examined complete results. However, it is too difficult to understand what the key point is the authors want to present. I have several comments as follow.

  1. The authors mentioned that Ze 450 can induce a metabolic shift from oxidative phosphorylation to glycolysis. Dose this effect facilitate to prolong lifespan? And why?
  2. In Figure 2d, I can’t find the no decreased protein expression after Ze450 treatment. Do you have any quantified data for this?
  3. On line 466-469, authors mentioned “estradiol-mediated protection against oxidative damage was independent of estrogen-receptor mediated effects, since only high micro molar concentrations of 17β-estradiol mediated protection, whereas concentrations of the estrogen relevant for receptor stimulation showed no effect.”. Please explain or provide the evidence to show why it was independent of estrogen-receptor mediated effects.
  4. On line 485, authors mentioned “Analyzing the rate of glycolysis the reduction of the ECAR mediated by erastin was partly prevented by 50 μM 17β-estradiol and Ze 450 (Figure A5 (d)).” Please check the labeled figure. These data were showed in Figure A5 (e) and (f) but not Figure A5 (d).
  5. As website indicated that The MTT assayis a colorimetric assay for assessing cell metabolic activity. NAD(P)H-dependent cellular oxidoreductase enzymes may, under defined conditions, reflect the number of viable cells present. Please explain why you mentioned that At 50 μM, 4-OH-tamoxifen exerted strong effects on metabolic activity (Figure A6 (d)) on line 498.
  6. Why the data of Metabolic activity (MTT assay) of 100ug/ml Ze450 treatment is different in Figure A6c and A6e?
  7. Did 10 μM 4-OH-tamoxifen combined with Ze450 have synergistic effect?
  8. In Figure 7a, 50 and 100 ug/ml showed similar effect on lifespan, why choose 100 ug/ml to do further exp.? Similar question, why choose 25mM Metformin to do further exp.? Maybe 50 or 100mM Metformin have better effects against PQ.
  9. In MTT assay, Ze450 seems to decrease cell survival; in survival analysis, Ze 450 seems to decrease the survival rate? Is Ze450 good and safe for using?
  10. On line 582, the authors mentioned that Metformin did not improve median survival (22 days M9 and 25mM metformin). But in Figure 7f, Metformin can prolong lifespan significantly.
  11. Why can the chemotactic index reflect neuronal function?

Author Response

Rebuttal letter to Reviewer 1

First of all, we thank the reviewer for careful evaluation of our manuscript and the encouraging detailed comments. We fully agree that the investigated effects of the cimicifuga extract Ze 450 on energy metabolism, cellular resilience and lifespan are complex and need appropriate introduction, presentation of the data and adequate discussion. We answered all questions raised by the reviewer and accordingly amended presentation and discussion of the data in the revised version of the manuscript. Please find the detailed answers to the reviewer’s questions below and according changes in the manuscript marked in red font as indicated in detail below.

  1. The authors mentioned that Ze 450 can induce a metabolic shift from oxidative phosphorylation to glycolysis. Dose this effect facilitate to prolong lifespan? And why?

This is a very important question and a key question for the understanding of the relevance of the observed effects in this study. There is in fact a lot of evidence in the literature for beneficial effects of the observed metabolic shift to many (age-related) diseases, metabolic diseases such as type 2-diabetes and ageing itself.  This is also mentioned in the manuscript, e.g. lines 640ff, 663ff, 701ff, and the related Warburg effect has been frequently linked to cellular resilience to stress, ageing, and to increased lifespan of whole organisms, as discussed also extensively for the effects of metformin (lines 701ff). The underlying mechanisms are only partly understood but there are –in short- two major mechanisms that are currently discussed: First, the metabolic shift to glycolysis allows for supply of fast energy – quite inefficient but fast which may be important in conditions of cellular stress, immune responses etc – i.e. the metabolic shift provides increased resilience to cellular stress in the ageing organism and age-related diseases. Here, we provide evidence for underlying mechanisms of this metabolic shift to enhanced glycolysis as presented in the results and discussed in lines 643 ff). Second, but maybe as important, reduced energy supply by aged and probably damaged mitochondria in the ageing organism will lead to increased mitochondrial ROS production thereby resulting in further cellular stress and damage to mitochondria in a vicious circle (discussed e.g. in lines 636 ff and summarized in lines 716 ff.). The damaged mitochondria are detrimental to the cells and organ functions and will activate pro-inflammatory responses – and these are mechanisms promoting ageing. Reduced mitochondrial respiration with concomitant energy supply through glycolysis allows for enough energy to keep cellular and organ function and reduced production of detrimental mitochondrial ROS production and ageing processes.

  1. In Figure 2d, I can’t find the no decreased protein expression after Ze450 treatment. Do you have any quantified data for this?

This observation is fully correct, there is no difference in the expression of the detected proteins in Fig 2d between Ze450-treated cells and the controls and the blot was quantified as presented in Figure A2b-f in the supplement. This is also stated in lines 404-405 in the manuscript: “Further, it was excluded that the inhibitory effect of Ze 450 was attributed to decreased protein expression (Figure 2 (d), Figure A2 (b-f))“. Following the reviewer’s notion this statement is now more clear since we added: “There were no differences detected between controls and Ze 450-treated cells regarding the expression levels of the proteins representing the mitochondrial respiratory complexes.

  1. On line 466-469, authors mentioned “estradiol-mediated protection against oxidative damage was independent of estrogen-receptor mediated effects, since only high micro molar concentrations of 17β-estradiol mediated protection, whereas concentrations of the estrogen relevant for receptor stimulation showed no effect.”. Please explain or provide the evidence to show why it was independent of estrogen-receptor mediated effects.

The remark by the reviewer is well taken and, of course, at high µM concentrations there will be also full activation of the estrogen receptor. However, the activation of estrogen receptor signaling cannot account for the observed protective effects of 17ß-estradiol. It is well established that the hormone 17ß-estradiol activates estrogen receptors at low nanomolar concentrations; the EC50 values are usually detected around 0.1 nM concentrations of 17ß estradiol, and full activation of these receptors is expected at concentrations of 17-ßestradiol at 1 nM. Thus, the concentrations applied here are already beyond the EC100 concentrations required for full estrogen receptor activation – and still no significant protective effect was observed at concentrations of 1-10 µM.

The anti-oxidative properties of estrogens are also well documented, and confirmed here, at concentrations of 17ß-estradiol higher than 10 µM – i.e. concentrations far beyond the concentrations needed for full estrogen receptor activation. In conclusion, 17ß-estradiol is only protective at concentrations where the antioxidative capacity is fully active but not at lower nanomolar concentrations that are required for activation of estrogen receptors. This allows for the conclusion that activation of the estrogen receptor was not involved in the observed protective effects against oxidative cell death. This is well in line with previous publications from our laboratory and by others demonstrating antioxidative capacities by estrogens independent of estrogen receptor signaling (e.g. Culmsee et al., Neuroprotection by estrogens in a mouse model of focal cerebral ischemia and in cultured neurons: evidence for a receptor-independent antioxidative mechanism. J Cereb Blood Flow Metab. 1999; 19:1263-1269. doi: 10.1097/00004647-199911000-00011).

Here, we compared the activities of 17ß-estradiol to Ze450 to demonstrate that in the current model system of oxidative cell death activation of estrogen receptors is not sufficient to provide protective effects. In combination with the other experiments presented in this study we demonstrate that Ze450 does not mediate its protective effects through activation of estrogen receptors. This is an important point since the activity of Ze450 against postmenopausal complaints has been frequently linked to estrogen-like activities and cimicifuga extracts were even declared as “phytoestrogens”. However, Ze450 does neither mediate its effects against oxidative cell damage nor any other effect against postmenopausal complaints through stimulation of estrogen receptors. Cimicifuga extract Ze450 is not a phytoestrogen and this is very important for mechanistic interpretation of the current results and other activities of Ze450, also beyond the scope of the present study.

  1. On line 485, authors mentioned “Analyzing the rate of glycolysis the reduction of the ECAR mediated by erastin was partly prevented by 50 μM 17β-estradiol and Ze 450 (Figure A5 (d)).” Please check the labeled figure. These data were showed in Figure A5 (e) and (f) but not Figure A5 (d).

We thank the reviewer for careful evaluation and detecting this error. We corrected the sentence in line 485 accordingly.

  1. As website indicated that The MTT assay is a colorimetric assay for assessing cell metabolic activity. NAD(P)H-dependent cellular oxidoreductase enzymes may, under defined conditions, reflect the number of viable cells present. Please explain why you mentioned that At 50 μM, 4-OH-tamoxifen exerted strong effects on metabolic activity (Figure A6 (d)) on line 498.

This notion by the reviewer is correct. The MTT assay is foremost a metabolic assay developed for detection of cell viability and proliferation. It is also correct that the assay is widely used for detecting cell death (i.e. reduced cell viability) under defined conditions such as erastin-induced ferroptosis. This needs, however careful validation e.g. through cell death assays as for example Annexin V/PI staining as also applied in the current study and in our previous studies and by others.

Importantly, however, reduced MTT signal does not always mean that the cells are damaged. For example, we showed in a previous study that Ze 450 at concentrations higher than 200 µM significantly reduced the metabolic activity in cultured cells as detected by the MTT assay, but did not induce cell death as detected by Annexin V/PI staining (Rabenau et al., 2018, PMID: 30519610; Rabenau et al., 2019, PMID: 30599889). Further, reduced metabolic activity and reduced cell proliferation as detected by the MTT assay or the cell impedance measurements, respectively, were fully recovered after Ze450 withdrawal further confirming that the cells were not damaged by Ze 450, but these effects were attributed to reduced metabolic activity.

It is also well observed that 50 µM 4-OH tamoxifen induced a strong decline of the MTT signal and we suppose that this actually reflected indeed toxic effects, i.e. cell death. Thus we excluded this concentration from further experiments but did not differentiate whether the observed decline of the MTT signal was attributed to reduced metabolic activity because of cell death. We clarified this and state now in line 500: “Here, the effects of 4-OH-tamoxifen were compared to Ze 450 in neuronal cells. At 50 µM, 4-OH-tamoxifen exerted strong reduction of metabolic activity, likely reflecting toxic effects and cell death at this high concentration (Figure A6 (d)),…”

  1. Why the data of Metabolic activity (MTT assay) of 100ug/ml Ze450 treatment is different in Figure A6c and A6e?

The difference mentioned by the reviewer is only a difference in the level of significance- i.e. in the experiment depicted in Fig A6e Ze 450-significantly reduced metabolic activity detected by the MTT assay, whereas this reduction was not reaching statistical significance in Fig A6c. The effects, however, are very similar and -if compared between the experiments- would not be different from each other. Overall these findings further support the notion that Ze 450 reduced the metabolic activity under basal conditions as detected by the MTT assay and confirmed in the assays directly measuring mitochondrial respiration.

  1. Did 10 μM 4-OH-tamoxifen combined with Ze450 have synergistic effect?

To answer this point, we stated in line 502 ff: “To further study synergistic or additive effects of Ze 450 with 4-OH-tamoxifen, low, non-protective concentrations of 4-OH-tamoxifen (0.5 and 1 µM) were used (Figure A6 (e)). The results show that no additional effects were detected in comparison to Ze 450 (Figure A6 (e))“ In the present study, we applied 4-OH-tamoxifen as a pharmacological tool, i.e. to block estrogen receptor signaling in order to support the conclusion that estrogen receptor activation was not involved in protective activities of Ze450. The results from our study do not support a synergistic interaction between tamoxifen and Ze450 and further analyses of the mechanism underlying the observed effects of tamoxifen are beyond the scope of the present study.

  1. In Figure 7a, 50 and 100 ug/ml showed similar effect on lifespan, why choose 100 ug/ml to do further exp.? Similar question, why choose 25mM Metformin to do further exp.? Maybe 50 or 100mM Metformin have better effects against PQ.

This is a valid question and we added a remark in line 581 to clarify that we used these concentrations in C. elegans since they matched the concentrations of Ze450 applied in the neuronal cells in vitro. For Metformin, 50 and 100 mM may indeed have even better results than 25 mM as suggested by the reviewer, but already 25 mM is cobsidered as a very high concentration which had to be applied for reasons of low bioavailability. However, concentrations higher than 25 mM may exert off target effects and are also far beyond corresponding doses required e.g. for complex I inhibition. Thus we decided to use concentrations that are also applied in vitro. Following the reviewer’s notion, we included the remark in line 581: “Following these promising findings, Ze 450 and metformin were further investigated for their effects on the lifespan of C. elegans in a real-time assay at concentrations matching the concentrations previously used in the cultured neuronal cells ...”

  1. In MTT assay, Ze450 seems to decrease cell survival; in survival analysis, Ze 450 seems to decrease the survival rate? Is Ze450 good and safe for using?

This is a very important point raised by the reviewer that was clarified in extensive previous studies. First of all: Ze 450 is safe for using also in patients in the clinic and this is very well validated in experimental and clinical studies and also by stringent and continuous revision of Ze 450 pharmacovigilance. Ze 450 is approved and registered in 14 countries. In Switzerland, the home market, Ze 450 was launched in 2005. Since the marketing authorization until 01-Aug-2020, according to the periodic safety update report (PSUR) from the producing company (Max Zeller Söhne AG, Romanshorn, Switzerland), 280 million defined daily doses (DDD) were sold with an overall reporting rate of 1 reported reaction for approximately 201 000 DDD.

As discussed before, the MTT assay is foremost an assay detecting cell metabolism and cell proliferation. Only under defined conditions and with adequate corresponding detection of cell death as also performed by AnnexinV/PI assays a reduction in the MTT signal can be interpreted as reduced cell viability or cell death. We have extensively investigated the phenomenon of reduced MTT signal/ metabolic activities by Ze 450 befor as published in (Rabenau et al., 2018, PMID: 30519610; Rabenau et al., 2019, PMID: 30599889) and also found in References 2, 12, 16, among others.

From these previous studies we know, as stated before, that the observed decline in the MTT signal fully recovers and the metabolic shift reverses after Ze450 withdrawal. We also know and presented in this study that the metabolic shift and reduced metabolic activity is a prerequisite for the protective effects by Ze450 against oxidative cell death. This was the reason to further investigate this phenomenon in more detail as presented in the present study including our findings on longevity.

  1. On line 582, the authors mentioned that Metformin did not improve median survival (22 days M9 and 25mM metformin). But in Figure 7f, Metformin can prolong lifespan significantly.

This is again a very important point highlighted by the reviewer. As mentioned by the reviewer we state exactly what was found and depicted in Figure 7f: “The results showed that Ze 450 prolonged the median survival (24 days) and the overall survival (Figure 7 (e)) compared to vehicle control (20 days). Moreover, 25 mM metformin demonstrated similar results regarding the overall survival rate compared to vehicle control (Figure 7 (d)), but metformin did not improve median survival (22 days M9 and 25 mM metformin). In this real-time setting, we demonstrated robust effects of Ze 450 on lifespan, which were more pronounced compared to metformin.” These findings confirm previous findings with metformin in this model system and also now reveal that Ze450 not only increased overall lifespan at very much lower concentrations than required for metformin, but also enhanced the medium lifespan- i.e. the effect was pronounced throughout the time of observation.

  1. Why can the chemotactic index reflect neuronal function?

As described in the reference [29], Margie, O.; Palmer, C.; Chin-Sang, I. C. elegans chemotaxis assay. J. Vis. Exp. 2013, e50069, doi:10.3791/50069], “C. elegans has impressive chemotaxis behavior. The premise behind testing the response of the worms to an odorant is to place them in an area and observe the movement evoked in response to an odorant”. C. elegans is used to investigate neuronal function, based on the nematodes’ well-developed sensory and olfactory system as stated in line 597 of the manuscript, and these are complex neuronal functions based on sensoric and motoric nerves. Thus, chemotactic behavior was studied to assess effects of Ze 450 on these neuronal functions along established protocols widely established in this field of research.

Reviewer 2 Report

The article elucidates several properties of a heterogeneous and partially purified extract obtained by a Cimicifuga racemosa, a plant widely used as an herbal medicinal product. The novelty of this paper is mean because the extract ZE450 has been already described in another paper, but the research is still interesting and deserves to be considered by the editorial board for publication in this journal, although the authors did not recognize the molecular identity/ies of the molecule/s responsible for the phenotypes described in the text. However, several points should be adequately addressed to have my enthusiastic support.

  1. Figures 1C, 4, and 5, the links between the bars should be included. It is too hard to understand the significance of the bars and the comparison.
  2.  (399) “In line with previous findings, Ze 450 affected pyruvate and malate driven respiration in a time-dependent manner, but did not completely inhibit complex I activity since the injection of rotenone”. This point should be at least exhaustively debated in the discussion, but a dose-dependent response, abrogating the OxPhos, will provide more strength to this sentence.
  3. Concerning figure 2, I cannot understand how FCCP (Port B) can remedy the effect of Rotenone (Port A). Rotenone blocks the Complex I contribution to the electron chain. This suggests that the Complex II is sufficient to sustain -in part- the respiration, but if it is true I cannot believe that “30 min” and “60min” treatments are so different. This might mean that ZE450 affects also the Complex II. I encourage the author to repeat the test, including two conditions Rotenine+a ComplexII inhibitor (e.g. TTFA), and the ComplexII inhibitor without Rotenone. 
  4.  Even more, could the author better discuss the concept of “time-dependent”? It is not clear if the effect is mediated to either the accumulation of a byproduct, or the progressive inhibition of the enzyme, or mitochondrial swelling. This point could be addressed by measuring the slopes, the basal, and the plateau levels in a seahorse experiment( analog of that fig.2C -Ctrl, 100ug/ml 30min,and 100ug/ml 60min). In these experiments, the authors should focus their attention on the time point “start to port A”, checking the action of the compounds until the signal is at the “plateau” phase. I am supposing that plateaus and slopes of the conditions 100ug/ml 30min and 100ug/ml 60min could provide new information.

Round 2

Reviewer 1 Report

No more comments.

Reviewer 2 Report

No more comments.